# The resilient potential behaviours in an Internal Medicine Department: Application of resilience assessment grid

Mariam Safi [1,2,3] *, Bettina Ravnborg Thude[1,2], Frans Brandt[1,2], Robyn Clay-Williams[3]

1 Internal Medicine Research Unit, University Hospital of Southern Denmark, Aabenraa, Denmark,
2 Department of Regional Health Research, University of Southern Denmark, Odense, Denmark,
3 Australian Institute of Healthcare Innovation, Macquarie University, Sydney, NSW, Australia

* Mariam.Safi2@rsyd.dk, safi_mariam@hotmail.com

## Abstract

### Background

The healthcare system is frequently subject to unpredictable conditions such as organisational changes and pandemics. In order to perform as required under these conditions (i.e. exhibiting resilient behaviour), it is necessary to know the current position of the organisation with respect to the four resilient potentials i.e. respond, monitor, learn and anticipate. The study aimed to understand and assess resilient performance of an Internal Medicine Department in a public hospital in Denmark using the resilience assessment grid (RAG).

### Methods

A modified Delphi method was used to develop the context specific RAG, using interviews to generate items, two rounds of expert panel reviews and pilot testing the developed RAG questionnaire. The four sets of structured RAG questions were tested and revised until satisfactory face and content validity for application was achieved. The final version of the RAG (28-item Likert scale) questionnaire was sent electronically to 87 healthcare professionals (clinicians and managers) in January 2021 and 2022. The data was statistically analysed and illustrated in radar charts to assist in interpreting the resilience profiles.

### Results

While the resilience profiles in 2021 and 2022 were similar, the scores in 2022 were slightly lower for some of the sub-indicators. The results indicate areas for improvement, especially related to the Internal Medicine Department's potential to respond and learn. The results from the RAG were presented to the chief clinical consultants and managers to identify initiatives for quality improvement and for planning a new workflow at the Internal Medicine Department.

**Data Availability Statement:** Data is available upon request to the first author or University Hospital of Southern Denmark (SHS. Forskning@rsyd.dk). There are legal restriction on

sharing data by the Region of Southern Denmark (case number 20/50532) cf. Art 30 of The EU General Data Protection Regulation. Data contain potentially personally-identifying information. It is a small data set and the identity of the respondents can potentially be connected.

**Funding:** This work was supported by the University Hospital of Southern Denmark as part of a Ph.D. project. The funders had no role in study design, data collection and analysis, decision to publish, or preparation of the manuscript.

**Competing interests:** All authors have no conflict of interest to declare.

## Conclusion

The RAG is a managerial tool to assess the potential resilient performance of the organisation in respect to the four resilience potentials, i.e., responding, monitoring, learning, and anticipating. It can be used to construct the resilience profile of the system over time to manage organisational changes.

## 1. Introduction

The use of non-linear models to understand and manage performance has gained popularity in the healthcare discourse [1]. Resilience engineering (RE) discipline provides a better understanding of the complexity of everyday work in a socio-technical system such as healthcare and focuses on human adaptive capacities for maintaining effective system performance-[2]. It acknowledges that acceptable and adverse outcomes in healthcare have a common basis, i.e., they result from everyday performance variability-[3, 4]. Resilience can be defined as a system's ability to perform as required under a variety of conditions which includes responding appropriately to changes, disturbances, and opportunities-[5]. Thus, resilience engineering is the ability to engineer systems that can withstand internal and external disturbances by adapting to new and unexpected circumstances. The resilience engineering perspective presents four core potentials that jointly enable resilient performance, i.e., responding, monitoring, learning, and anticipating-[6]. The potential to respond considers what an organisation needs to respond timely and effectively to what happens-[5, 7]. The potential to monitor looks at how well the organisation monitors its operations and detect changes to work conditions that may affect the day-to-day work-[5, 7]. The potential to learn looks at the degree to which learning is integrated into the organisation and whether the organization learns from past experiences-[5, 7]. Lastly, the potential to anticipate addresses the organisation's ability to predict what may happen in the future beyond the range of current operations and their effect on the organisation-[5, 7]. To what extent these resilient potentials are present (or absent) in the organisation determines its ability to perform resiliently-[5, 7]. The four potentials for resilient performance, translated into the resilience assessment grid (RAG)-[5, 7] developed by Hollnagel, can be used as proxy measures to assess the organisation's ability to perform resiliently. The RAG framework consists of a generic questionnaire that can be used to construct a resilience profile of the organisation over time, in terms of the four potentials of resilience. The RAG needs to be tailored to each new system in which it is applied and be used by a single system repeatedly for continuous improvement and comparison-[5, 7]. The RAG does not provide a guide for how to develop the questions. It is a pragmatic tool that puts emphasis on the questions being operational, convenient and easy to develop. Additionally, Hollnagel [5] states that the questions should be answered by a small and stable group of respondents involved in the activities to ensure the reliability of the assessments. A strength of the RAG is that it has been applied widely across several domains such as healthcare [3, 8], aviation [9], traffic management [10], nuclear plant [11] and water sector [12] to analyse and support resilient performance. In healthcare, the RAG has been primarily applied in emergency care [8, 13–15] and anesthesia departments [3, 16]. Compared to other tools measuring resilience potentials such as the Relative Overall Resilience (ROR) [17, 18], the RAG has a broader scope and covers the capabilities of resilient performance to assess overall system performance. The RAG can be used to support organisational changes before, during and after the implementation [5]. However, previous studies on RAG have primarily focused on developing the RAG and it is not

clear how RAG was used to improve performance in practice [8]. In the current study, we investigated how the resilience profiles can be used to support organisation structural changes in practice.

The study aimed to understand and assess the resilient performance of the Internal Medicine Department using the RAG method. Specifically our intention with using RAG was:

- To determine the baseline resilience profile or position of the Internal Medicine Department.

- To determine if the baseline position was maintained during organisation structural changes.

- To repeat the RAG and use the comparative results for planning quality improvement initiatives.

The Internal Medicine Department consist of different specialist outpatient clinics and is a representation of other healthcare facilities. Our study, may guide similar practices or other industries in how to use the RAG to construct resilience profile and support structural organisation changes over time.

## 2. Methods

We employed a mixed-method study combining qualitative and quantitative methodology. The study approach had three parts: 1) A modified Delphi method [19] was used to generate a tailored set of RAG questions adapted to the Internal Medicine Department; 2) The tailored RAG questionnaire was applied in a survey format; and 3) Results were presented to the chief clinical consultant and the managers. See Fig 1. to gain an overview of the timeline.

### 2.1 Setting: The internal medicine department

In Denmark, outpatient specialist care is delivered at hospital-based outpatient clinics for non-acute patients. The specialist can also refer patients to other specialists for diagnoses and treatment [20]. The outpatient care clinic is an example of a complex socio-technical system, meaning that the outpatient care clinic's performance and behavior emerge from the interaction

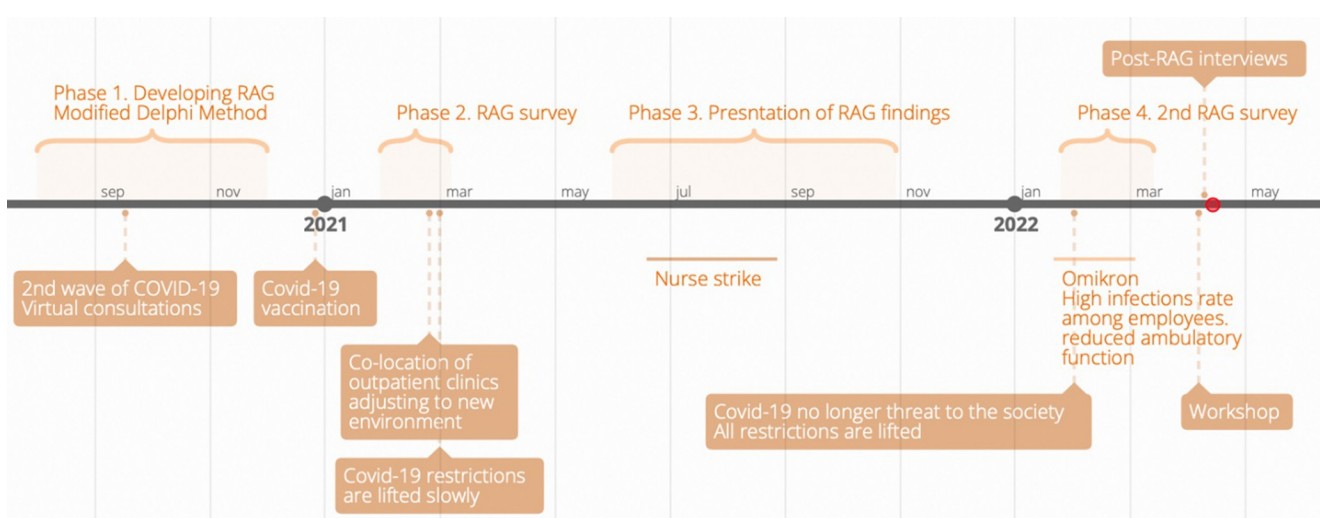

**Fig 1. Timeline of the study.**

between the people in the system and the technical artefacts (e.g., machines, governing bodies, and technique) [3]. The outpatient care has numerous stakeholders, each with different roles and interests, such as the frontline clinicians, managers, patients, and administrative employees. The combinations of interactions between these multiple stakeholders means that the outpatient clinic response is not consistent. Any disturbances or changes in the system or organisation can affect its ideal operating conditions and create variability in everyday performances.

Like most countries, specialist outpatient clinics in Denmark need to adapt their functioning under different conditions such as economic pressure [21], increasing volume of patients [22], demands from the governing bodies [21, 23] and structural re-organisation of hospitals [24], nurse strikes [25] and crises such as the unprecedented covid-19 pandemic. Similarly, the Internal Medicine Department at the University Hospital of Southern Denmark has been placed under growing strain because of covid-19 and had to find new solutions and workflows to adapt to the changing conditions. The Internal Medicine Department at the University Hospital of Southern Denmark consists of three specialist outpatient clinics; pulmonology-, nephrology- and endocrinology outpatient clinic. These clinics share a joint leadership and often have mutual patients. Amid the covid-19 pandemic, the Internal Medicine Department underwent considerable structural changes, such as co-location of the internal medicine outpatient clinics under the same roof to improve the patient pathway and the everyday activities of the outpatient clinics. Since the Internal Medicine Department was undergoing these changes, we thought it necessary to use the RAG to understand how we could support activities that enable the department to perform in a *resilient* manner.

## 2.2 Phase 1: Developing the RAG questionnaire using modified Delphi method

Fig 2. illustrates the RAG questionnaire development process.

A modified Delphi technique [19] as used to develop the context specific RAG, using interviews to generate items, two rounds of expert panel reviews and pilot testing the developed RAG questionnaire.

**Interviews and item generation.**   After an in-depth discussion with the RAG's developer, Erik Hollnagel, about the RAG framework, a semi-structured interview guide [26] (S1 Appendix) was developed based on the four resilience potentials.

Participants for the interviews were identified through purposive and snowball strategies with the assistance of the staff working at the hospital. We conducted 12 individual interviews and 3 focus group interviews. The focus group interviews consisted of 4–5 nurses and the duration of each interview was one hour. The individual interviews were conducted with participants in all layers of the department including physicians, middle managers, and chief consultants. The interviews were audio recorded and the pool of the potential interviews was guided by data saturation [27].

A theoretically informed thematic analysis [28–30] of the interview transcripts were undertaken. It combined an inductive and deductive approach for analysing the data. In the first stage of the analysis, an inductive data-driven process was used to identify the overarching themes. The researcher transcribed all the audio data. The researcher took notes and highlighted the text during transcription to document any ideas about themes. The transcripts were transferred to NVivo [12] software program and the researcher read each transcript in more detail to understand participants' responses on how they managed the complexity of their everyday clinical work. A tentative thematic structure was identified. In the second stage, the themes were further reviewed deductively informed by the RAG framework in terms of the four resilience abilities, i.e. learning, monitoring, anticipating, and responding. During this

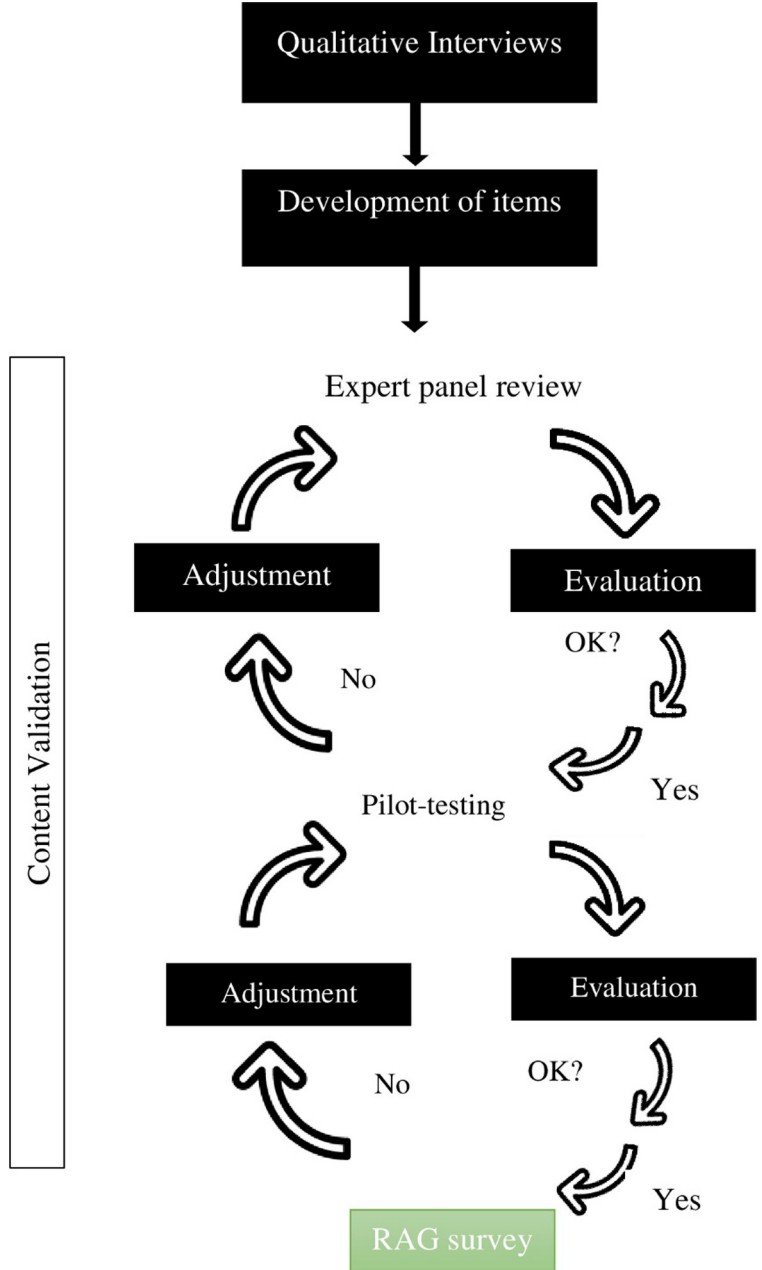

**Fig 2. The RAG questionnaire development process.**

process the transcripts and the themes were checked by a second author to ensure the credibility of the findings. This iterative process allowed for a more comprehensive analysis of the data than a simple descriptive analysis could provide.

The qualitative themes based on the four resilience potentials guided the item generation for the RAG questionnaire. The first author developed the initial questions according to the four abilities with the assistance of one of the authors with expertise in RAG.

**Expert review panels.** Two expert panels reviewed the items in order to achieve face and content validity [31, 32]. The first joint expert panel consisted of a resilience engineering

expert, one chief nursing leader, and one chief physicians' leader. The participants were provided with a paper format of the items and were asked in-person to review each item and rate it according to a four-point Likert scale from 'not relevant = 1' to 'highly relevant = 4'. While rating the items, the experts were interviewed to probe what they thought was meant by each questionnaire item to evaluate the comprehensiveness, comprehensibility, and relevance [31]. The researcher took notes during this process. The focus of the first round was on the 43 items–identifying duplications, which ones needed to be discarded or revised, and making efforts to determine if aspects of the construct were represented by items in correct proportion. This led to a reduction in items to 37. In the second round of expert review (n = 4), the revised set of items were administered to a new expert panel consisting of two physicians and two nurses to rate each item. The data was transferred to Microsoft Excel Version 2016. Then for each item, a content validity index (I-CVI) was computed as well as the overall scale content validity index(S-CVI) guided by the framework developed by Polit et al. [32, 33]. This lead to 28 items. For a scale to have excellent content validity, it would be composed of items that have I-CVI of 0.78 or higher and a S-CVI/AVG of 0.90 or higher. I-CVI somewhat lower than 0.78 were candidates for revision. The I-CVI was used to reduce the size of the questionnaire, using the proper terms and rephrasing some of the items to make them clearer and avoid misunderstanding. The final S-CVI/AVG was 0.97.

After the second round, the revised version with 28 items was sent to the developer of the RAG framework (Hollnagel), who critically reviewed its concordance with the underlying RAG construct. Lastly, the Chief of the Internal Medicine Department reviewed the RAG again.

**Pilot testing.** Five participants including nurses (n = 2) and physicians (n = 3) pilot tested the online RAG survey. The link for the survey was sent to them and they were asked to complete it. The participants completed the survey while the researcher was on the phone and took notes. The participants were encouraged to "think aloud" and verbalise their thoughts [31].To assess usability of the survey, the participants were asked about the layout of the questionnaire in Red Cap, and were timed when completing the survey. It took between 10–12 min to complete the RAG survey. Based on the minor feedback from the pilot testing the final version of the RAG survey was developed.

## 2.3 Phase 2: The application of the RAG questionnaire

**Data collection.** The final 28-items RAG survey (S2 Appendix) had two demographic questions asking about the participant's role and their work experience. At the end of the survey, there was a free form comment section. The survey consisted of eight items about responding, six items about monitoring, six items about learning, and five items about anticipating. The participants rated each question on a 5-point Likert scale (1–5) (e.g.,"*In the department, I know when my colleagues are under pressure and need help.*") (e.g., never, rarely, sometimes, often or always).

The online RAG survey was distributed twice, once in January 2021(n = 45) and again in February 2022 (n = 42) to nurses, physicians, and middle managers in the Internal Medicine Department. Medical and nursing students were excluded because they are at the department for a very short period of time. Reliable assessment is ensured by having a stable group of respondents that are involved in the activities that are being measured. According to Hollnagel [5], the number of respondents should not be too large and it should be easy and convenient to answer the questions. Reminder emails were sent one and two weeks after the survey opened and in the regular workplace briefing meetings, the participants were encouraged to fill out the survey. The survey remained available for a total of four weeks. Of note, the survey was conducted during the covid-19 period.

**Analysis.** The data from RedCap was transferred to STATA 17 [34]. Descriptive statistics were used to calculate the; a) mean score of each item; b) overall mean score for each of the four resilience abilities; and c) mean scores for each sub-group in the survey population. The Likert style data were treated as ordinal. The calculated means were transferred to Microsoft Excel Version 2016 to present the results in radar charts. The results are summarised and presented in tables and radar charts. This allowed for an in-depth interpretation of the data.

## 2.4 Phase 3: Presentation of the results

After the first round of the RAG survey, results were presented in three meetings to the chief clinical consultants of the department, managers (nurses and physicians with managerial roles) and nurses. In these meetings the first author presented and facilitated the meetings while a second researcher took notes that were summarised immediately afterwards. After the second RAG survey round, the results were presented to the chief consultants to elicit an understanding of the changed RAG scores from 2021.

## 2.5 Ethical considerations

The study was conducted according to the guidelines of the Declaration of Helsinki and was reported to and approved by the Region of Southern Denmark and listed in the internal record (case number 20/50532) cf. Art 30 of The EU General Data Protection Regulation. Oral and written information was provided to potential participants prior to inclusion. Written consent was obtained from all participants and participants were advised that they could recall their consent at any time without penalty.

## 3. Results

The response rate was 97.7% in 2021 and 81.8% in 2022, with 36 out of 44 of the respondents represented in the survey. Table 1 shows that the respondents are evenly distributed between nurses and physicians. Middle managers were the least represented in 2022 since 50% (3 out of 6) of the middle managers had not completed the survey. More than 50% of the respondents had over 5 years of experience.

Table 2 present the calculated mean scores and Fig 3 illustrates the resilience profile of the Internal Medicine Department in 2021 and 2022. Fig 4 presents the radar charts stratified by respondent groups, nurses, middle managers and physicians.

Table 1. Characteristics of study population.

| | | Year | Year |
|---|---|---|---|
| | | **2021** | **2022** |
| **Total** | | **N = 44 (100%)** | **N = 36 (100%)** |
| Function | Middle manager | 5 (11%) | 3 (8%) |
| | Nurse | 19 (43%) | 18 (50%) |
| | Physician | 20 (45%) | 15 (42%) |
| Work experience (years) | 0–1 | 3 (7%) | 2 (6%) |
| | 1–3 | 7 (16%) | 7 (19%) |
| | 3–5 | 6 (14%) | 7 (19%) |
| | 5–10 | 11 (25%) | 10 (28%) |
| | 10+ | 17 (39%) | 10 (28%) |

**Table 2. Overview of the mean scores for each item in 2021 and 2022 and stratified by respondent groups.**

| | Year | | | | Year | | | |
|---|---|---|---|---|---|---|---|---|
| | 2021 | | | | 2022 | | | |
| Respondent group | All | Middle Manager | Nurse | Physician | All | Middle Manager | Nurse | Physician |
| Total respondents (N) | N = 44 | N = 5 | N = 19 | N = 20 | N = 36 | N = 3 | N = 18 | N = 15 |
| **Respond total mean (SD)** | 2.82(0.35) | 2.97(0.41) | 2.82(0.28) | 2.79(0.40) | 2.66(0.42) | 2.79(0.19) | 2.68(0.30) | 2.60(0.5) |
| *R1_Flexibility*: There is time flexibility in my ambulatory program. | 2.1 (0.9) | 2.4(1.5) | 2.2 (0.6) | 2.0(1.0) | 1.9(0.7) | 1.7(0.6) | 2.1(0.6) | 1.6(0.7) |
| *R2_Teamwork*: In the department, we help each other in stressful/ under pressure situations | 3.2(0.8) | 3.2(0.4) | 3.4(0.6) | 3.0(0.9) | 3.1(0.9) | 3.3(0.6) | 3.1(0.9) | 3.0(0.9) |
| *R3_Leveraging Knowledge*: In the department, we can handle/ undertake each other's functions within the same professional group. | 2.8(0.8) | 3.4(0.5) | 2.4(0.6) | 3.1 (0.8) | 2.8 (0.8) | 2.3(0.6) | 2.6(0.8) | 3.3(0.7) |
| *R4_Shared priorities*: In the department, we have a common understanding of what we should prioritise. | 3.2(0.5) | 3.0(0.7) | 3.3(0.5) | 3.2(0.5) | 3.0 (0.6) | 2.7(0.6) | 3.2(0.4) | 2.9(0.8) |
| *R5_Ressources*: In the department, we plan with the right number/ amount of human resources to be able to perform everyday tasks. | 3.1(0.6) | 3.4(0.9) | 3.2 (0.4) | 3.0(0.6) | 2.8(0.8) | 3.7(0.6) | 2.8(0.9) | 2.5(0.7) |
| *R6_Self-managed*: In the department, we are self-managed and can handle daily operations without a daily manager. | 2.9(0.9) | 2.8(0.8) | 3.1(0.4) | 2.7(1.1) | 2.8 (0.6) | 2.7(0.6) | 2.9(0.3) | 2.8(0.8) |
| *R7_ Interruptions*: In the department, I do not experience many interruptions in the everyday work that prevent me from being able to perform my work | 2.2(0.8) | 2.4(0.5) | 1.9(0.6) | 2.3(1.0) | 2.0(0.8) | 2.3(0.6) | 2.2(0.9) | 1.7(0.8) |
| *R8_staff engagement*: In the department, we are motivated to solve tasks across specialties. | 3.1(0.7) | 3.2(0.8) | 3.1(0.8) | 3.0 (0.7) | 2.9 (0.8) | 3.7(0.6) | 2.7 (0.6) | 3.0(0.9) |
| **Monitor total mean (SD)** | 3.09(0.48) | 3.39 (0.51) | 3.16(0.31) | 2.95(0.58) | 2.96(0.50) | 2.89(0.67) | 3.04(0.39) | 2.88(0.59) |
| *M1_role and responsibility*: In the department, I know what my colleagues are doing and what their competencies can be used for. | 3.4(0.6) | 3.6(0.5) | 3.5(0.5) | 3.3(0.6) | 3.1(0.5) | 3.0(0.0) | 3.3(0.6) | 2.9(0.5) |
| *M2_Communications*: In the department, we communicate with each other to ensure that we solve the tasks. | 3.1(0.8) | 3.6(0.5) | 3.3(0.6) | 2.8(0.9) | 3.1(0.7) | 3.3(0.6) | 3.2(0.6) | 2.9(0.7) |
| *M3_Situation awareness*: In the department, I know when my colleagues are under pressure and need help. | 2.8(0.7) | 3.0(0.7) | 2.8(0.4) | 2.6(0.8) | 2.8(0.7) | 2.7(0.6) | 2.9(0.5) | 2.7(1.0) |
| *M4_Evaluation*: In the department, we are aware of continuously improving workflows. | 2.9(0.7) | 3.2 (0.8) | 2.9(0.4) | 2.7(0.9) | 2.7(0.8) | 3.0(1.0) | 2.8(0.6) | 2.5(0.9) |
| *M5_Organisational support*: In the department, we have the opportunity to get an overview of the day's work tasks. | 3.1(0.8) | 3.6(0.5) | 3.1(0.7) | 3.0(0.8) | 2.9(0.9) | 2.7(1.5) | 3.0(0.8) | 2.9(1.0) |
| *M6_Leadership*: In the department, I can easily get in touch with my immediate manager. | 3.3(0.7) | 3.5(0.6) | 3.3(0.7) | 3.3(0.8) | 3.1(0.6) | 2.7(0.6) | 2.9(0.5) | 3.3(0.7) |
| **Learn total mean (SD)** | 2.94(0.47) | 3.00(0.37) | 2.90(0.41) | 2.97(0.56) | 2.75(0.49) | 2.94(0.54) | 2.71(0.54) | 2.77(0.44) |
| *L1_ Knowledge Dissemination*: In the department, we share relevant professional knowledge. | 2.9(0.7) | 3.0(0.7) | 3.1(0.6) | 2.8(0.7) | 2.9 (0.7) | 2.3(0.6) | 2.8 (0.6) | 3.1(0.7) |
| *L2_safety culture*: In the department, I feel safe asking about something I do not know. | 3.7(0.5) | 3.6(0.5) | 3.6(0.5) | 3.8(0.4) | 3.5(0.8) | 3.3(0.6) | 3.7(0.5) | 3.3(1.0) |
| *L3_relevance*: I get useful answers to my questions. | 3.3(0.6) | 3.6(0.5) | 3.4(0.5) | 3.2(0.7) | 3.4(0.5) | 3.7(0.6) | 3.4(0.5) | 3.3(0.5) |
| *L4_Development*: In the department, I have sufficient support to develop or improve myself (through new work assignments, training, education, increased responsibility, etc.) | 3.0(0.8) | 3.2(0.8) | 2.8(0.7) | 3.0(0.9) | 2.7(0.9) | 3.0(1.0) | 2.6(1.0) | 2.7(0.7) |
| *L5_Learning from what goes well*: In the department, we use our experiences from good patient cases to learn. | 2.5(0.8) | 2.4(1.1) | 2.4(0.8) | 2.8(0.9) | 2.3(1.0) | 3.0(1.0) | 2.1(1.1) | 2.5(0.8) |
| *L6_Feedback process*: In the department, we have sufficient time to follow up on efforts and learn from it. | 2.2(0.7) | 2.2(0.4) | 2.1(0.7) | 2.3(0.8) | 1.8(0.7) | 2.3(0.6) | 1.7(0.8) | 1.7(0.6) |
| **Anticipate total mean (SD)** | 2.83(0.45) | 2.96(0.67) | 2.77(0.38) | 2.85(0.47) | 2.69(0.49) | 3.00(0.40) | 2.54(0.47) | 2.80(0.49) |
| *A1_Expertise*: In the department, we have the competencies needed to carry out our work. | 3.3(0.5) | 3.6(0.5) | 3.3(0.5) | 3.3(0.5) | 3.2(0.6) | 3.0(0.0) | 3.2(0.5) | 3.2(0.7) |

*(Continued)*

**Table 2.** (Continued)

| Respondent group | Year 2021 | | | | Year 2022 | | | |
|---|---|---|---|---|---|---|---|---|
| | All | Middle Manager | Nurse | Physician | All | Middle Manager | Nurse | Physician |
| Total respondents (N) | N = 44 | N = 5 | N = 19 | N = 20 | N = 36 | N = 3 | N = 18 | N = 15 |
| A2_Valnerable: In the department, we are aware of where we have challenges. | 2.9(0.6) | 2.8(0.8) | 2.7(0.5) | 3.0(0.6) | 3.0(0.6) | 3.0(0.0) | 2.9(0.6) | 3.1(0.6) |
| A3_Opportunistic mindset: In the department we have focus on identifying future opportunities | 2.6(0.6) | 2.8(0.8) | 2.5(0.5) | 2.5(0.6) | 2.4(0.8) | 3.0(1.0) | 2.3(0.8) | 2.3(0.7) |
| A4_Proactiv: In the department, we work actively to improve our work with a view to future challenges and requirements. | 2.8(0.6) | 3.0(0.7) | 2.7(0.6) | 2.8(0.7) | 2.4(0.9) | 3.0(1.0) | 2.2(1.0) | 2.6(0.6) |
| A5_Communication: In the department, plans are clearly communicated to staff. | 2.6(0.7) | 2.6(0.9) | 2.6(0.6) | 2.5(0.8) | 2.4(0.9) | 3.0(1.0) | 2.1(0.9) | 2.8(0.8) |

\*\*\*Data is presented as mean (SD)

## 3.1 The potential to respond

The Internal Medicine Department's potential to respond was assessed using eight sub-indicators. The radar charts showing the respond potential from 2021 and 2022 are very similar. In both 2021 and 2022, the department scored low in *R1_flexibility* in ambulatory programs and *R7_interruptions*, meaning that the respondents experienced interruptions during their workday which may have prevented them from being able to perform their work. It is relevant to notice that the department scored high in *R5_ressources*, meaning that the department planned for the right amount of human resources to be able to perform everyday tasks. Even though the patterns in the radar charts are very similar, we can see that the scores for 2022 were slightly lower.

We stratified the data by respondent groups to gain a more nuanced overview of the department's performance. In both years, the nurses' scores for *R3_leveraging knowledge* differed from the overall scores and from the other respondent groups. However, *R3_leveraging knowledge* improved a little in 2022 (2.6), meaning that the nursing teams were more likely to undertake each other's functions in 2022 compared to 2021. In contrast, the *R3_leveraging knowledge* (2.3) decreased for middle managers in 2022. Additionally, physicians rated lower in *R7_interruptions* (1.7) in 2022 compared to 2021(2.3), whereas the nurses (2.2) improved their scores in 2022 compared to 2021(1.9).

## 3.2 The potential to monitor

The Internal Medicine Department's potential to monitor was assessed using six sub-indicators. While the department scored a little bit lower in *M1_role and responsibility(3.1)*, *M4_evaluation(2.8)* and *M4_organisational support(2.7)* and *M6_leadership (3.1)*, it maintained its position in *M2_communication* (3.1) and *M3_situation awareness(2.8) (see Table 2)*.

In the stratified radar charts (Fig 4), the middle managers scored a little lower in all of the sub-indicators compared to 2021. According to the data, the physicians (2.7) and nurses (2.9) had slightly improved scores *for M3_situation awareness*, meaning that they were often aware of when their colleagues were under pressure and needed help. This is aligned with *M2_communication* scores (3.1); the healthcare professionals often communicated with each other to solve everyday tasks.

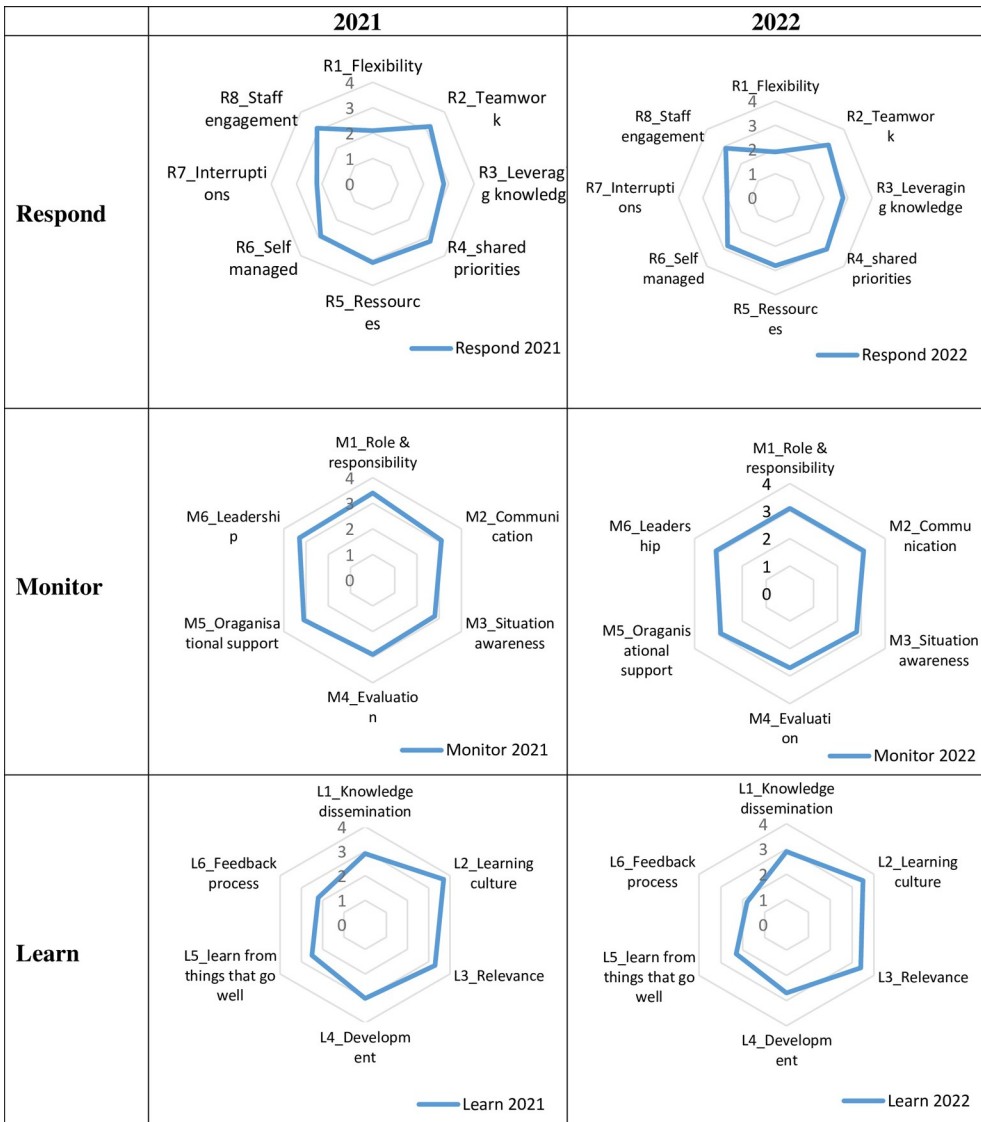

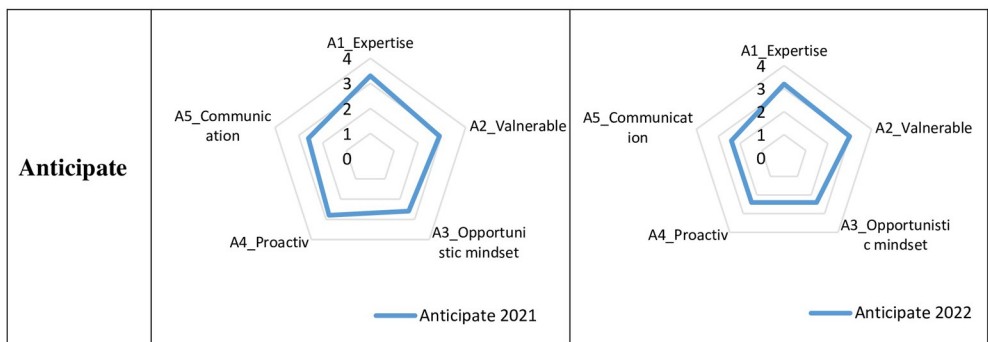

**Fig 3. Resilience profile of the internal medicine department in 2021 and 2022.**

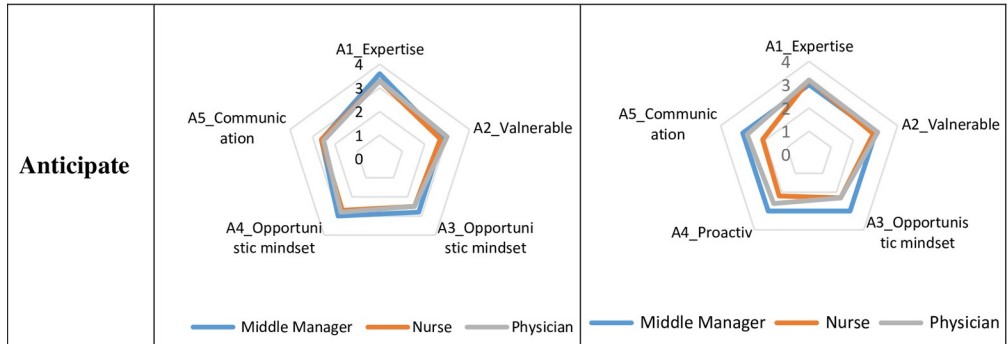

**Fig 4. Resilience Profile in 2021 and 2022 stratified by respondent groups.**

### 3.3 The potential to learn

The Internal Medicine Department's potential to learn was assessed using six sub-indicators. The radar chart from 2021 and 2022 followed the same pattern. Table 2 indicates that the potential to learn is skewed to the right. This means that the internal medicine department emphasised *L1_knowledge dissemination*, *L2_learning culture*, *L3_relevance* and *L4_development* in everyday clinical work to prepare their resilience potential. While some of the sub-indicators, *L5_learning from things that go well(2.3)* and *L6_fedback process(1.8)*, scored low in 2022, the department maintained its position overall.

The stratified radar charts from 2021 and 2022 respectively followed the same pattern as the overall score. While the nurses and physicians maintained more or less the same position as 2021, the middle managers rated higher in *L5_learn from things that go well* in 2022 (3) than in 2021(2.4).

### 3.4 The potential to anticipate

The Internal Medicine Department's potential to anticipate was assessed using five sub-indicators. While the pattern for the radar charts in 2021 and 2022 are similar, the scores from 2022 were a little bit lower. However, the radar charts illustrate that the Internal Medicine Department emphasised *A1_Expertise* and *A2_Valnerable*. This means that the department more than often had the competencies needed to perform the required tasks in everyday work and that they were aware of their challenges.

In the stratified radar chart, the middle managers improved scores in the sub-indicators compared to 2021. While the nurses maintained *A1_Expertise(3.2)* and improved their score in *A2_Valnerable(2.9)*, meaning that they were often aware of their challenges. Similarly, the physicians had improved scores in *A2_Valnerable(3.1)* and including, *A4_proactiv(2.6)* i.e. working actively to improve their work to face future demands and requirements.

### 3.5 Presentation of the results to staff

After the first RAG survey, the results were presented in three meetings to chief consultants, managers (nurses and physicians with managerial roles) and nurses. This allowed us to gain an in-depth understanding of the survey results and facilitate discussions on managerial actions to improve everyday work. The participants at the meetings were satisfied with the results and confirmed that they observed a similar pattern as illustrated in the radar charts. For example, regarding *Flexibility (R1)*, the participants confirmed that the lack of flexibility in their ambulatory program didn't allow for consultation with colleagues regarding patients. The healthcare professionals experienced *interruptions (R7)* while undertaking their care activities from telephone, communications regarding patients, etc. However, the majority of healthcare professionals considered the interruptions part of the clinical environment that was unavoidable. Nevertheless, everyone agreed that it was something that they would like to reduce moving forward. Further, they observed that the nurses scored lower in *leveraging knowledge (R3)*, the ability to undertake a colleague's functions or role. This was in relation to the shortage of staff because of covid-19 illness. For example, if one of the nurses was sick, the others were not able to fill that role because they didn't have the competencies and, additionally, were burdened with work from their own ambulatory program. They also commented that the nurses in the outpatient clinics were highly specialised in their own area, which makes it challenging to take on others' tasks or work functions. Overall, the participants primarily focused on improving the potential to respond.

## 4. Discussion

We applied the RAG method to the Internal Medicine Department, to evaluate its performance from a perspective of organisational resilience. The RAG method provided us with knowledge about the initial resilience state or position of the Internal Medicine Department and knowledge about how to maintain or improve the position. While the pattern of the radar charts from 2021 and 2022 were very similar, the mean scores show that the scores for 2022 were slightly lower. However, the stratified data by nurses, physicians and middle managers illustrated a more nuanced picture of the performance of the department. The stratified data showed that the different respondent groups had improved their performance in some of the sub-indicators even though the overall scores indicated the opposite. *Leveraging knowledge (R3)* improved for nurses in 2022, this may be due to an increased interest in ensuring that there are at least 2–3 nurses that can perform the tests. *Situation awareness (M3)* and *vulnerable*(A2 i.e. knowing where you have challenges) improved for nurses and physicians. An explanation may be that the healthcare professionals consistently communicated for example in their morning conferences. We can see that they had maintained their position in *M2_communication* with high scores 2.8. At the post RAG interviews, the Chief consultant of the Department suggested that an explanation for the lowered score in 2022 may be that the Internal Medicine Department was under a lot of pressure, including a structural re-organisation, covid-19 and the nurses' strike. Hence, the department did not have the time and resources to implement specific quality initiatives alongside the co-location of outpatient clinics. Additionally, the chief consultant observed that the clinicians are currently feeling more stressed than during covid-19, because the outpatient clinics had scaled back their work. This has resulted in post covid-19 lag such as long waiting lists, which is putting a lot of pressure on the healthcare professionals and the managers.

The Internal Medicine Department's goal was to maintain its position in 2021 and organise a quality improvement workshop when the covid-19 restriction was lifted. Our intention with the RAG was to keep a track of how the position changed over time, and to determine whether the change went in the desired positive direction [5]. The radar chart profiles from 2021 and 2022 showed that the Internal Medicine Department performed as *required* under varying circumstances and avoided major destabilising circumstances. Additionally, the RAG provided a good basis for developing mutual strategies among the outpatient clinics at the Internal Medicine Department to strengthen the collaborative work around patient care.

Previously there was no specific guideline for when to repeat the RAG. However, in the new white paper on Systematic Potential Management (SPM)(the new label for RAG) [5], Hollnagel clarifies that the time between the surveys depends on how rapidly (or slowly) changes take place. The frequency of RAG application should match the rate of organisational changes being considered–either deliberately or due to external conditions. Hollnagel suggests that a reasonable time interval could be between 6–12 months [5]. A solution may be to apply the RAG pre- and post-organisational changes. To our knowledge only the study by Hunte & Marsden [15] repeated their context specific RAG in their monthly interdisciplinary meetings. Repeating the RAG may be impacted by financial and other constraints on the project, such as funding and time. Furthermore, in previous studies on the RAG in healthcare [8, 14, 16] and other industries [11, 12] it was not clear if the outcomes of the RAG were actually used to drive practice improvement. Future studies should explore this.

A concern that has been expressed regarding the RAG is that it does not prescribe weighting of the four potentials. According to Hollnagel, it is important that the four resilience potentials are present to some extent in the organisation to achieve resilience performance [5]. Apneseth et al. [35] argues that the relative importance of the four abilities of RAG varies

depending on the system in question. Hence it is not necessary for all organisations to put the same emphasis on, for example, monitoring and anticipating ability in order to be resilient [35]. This is supported by the imbalance in the number of items for each of the four resilience potentials in many of the RAG studies. In our study, the emphasis naturally emerged on the respond potential because both the managers and the healthcare professionals found it important for their work and wanted to improve *R1_Flexibility in ambulatory program*, *R7_Interruptions* and *R3_leveraging knowledge*. This was identified both in the development phase of the RAG questionnaire and in post-survey presentation of results. Healthcare organisations tend to emphasis on 'response performance' because they operate as a response to patient demand (i.e. that's what they measure) [36, 37]. An alternate approach to weighting may be to use the analytical hierarchy process framework proposed by Patriarca et al. [16] to study the different weights of the abilities and their relative influence on overall organisational resilience.

In addition, it is important that the RAG survey is answered by a stable group of respondents and people in different organisational positions who are knowledgeable about the organisational functions, because the questions refer to something that is part of the respondents' competence or experience [5, 7]. We therefore found that not all questions pertaining to the four resilience potentials should be answered by the same group of respondents. An option may be to distribute the set of questions based on the respondents' position and perceived knowledge. From our own experience, for example, the RAG questions developed about the potential to anticipate were not relevant for the frontline healthcare professionals since they referred to activities that are part of the chief manager's and middle-manager's area of knowledge.

An advantage of the RAG is its flexibility and easy to read graphical results (resilience profile), as the organisation's weaknesses and strengths are easily understood. Similar to our study some studies on the RAG included sub-dimensions or themes such as leadership [16], social interaction [38] and sensemaking [39] as facets to better describe the underlying construct of each potential. It is also possible to include a fifth resilience category if it is relevant to the setting. The study by Darrow [39] included a fifth category, i.e. individual resilience. This indicates the RAG's flexibility and its increasing potential benefits to cover all aspects of the sociotechnical system. A recently introduced resilience measurement tool, the Relative Overall Resilience (ROR) [17, 18] also creates a resilience profile of a system; however, the ROR is generally focused on the response capability and used in manufacturing industry. In contrast, the RAG has a broader scope and is highly adaptable to assess the resilient performance of a healthcare organisation.

## 4.1 Implications for practice and future research

The Internal Medicine Department at The University Hospital of Southern Denmark in our study is a typical representation of hospitals on an international level that face similar challenges of increasing demands and limited resources. Our study could help guide similar practices, and hospital systems nationally and internationally, in how to develop and apply the RAG in their organisations to support organisational changes. Specifically, other Danish Hospitals could adapt our Danish RAG version to their organisation to assess its resilient performance. Furthermore, healthcare managers can use the RAG as a supplement to other managerial tools to initiate discussions on how the care of patients within a single hospital system is best organised. The hospitals in the Region of Southern Denmark primarily use Virigina Mason's Lean methods [40] for quality improvement in Hospitals. Future research should investigate how the RAG method could supplement Lean and other tools in hospitals. This may solve the challenges of using the RAG as a managerial tool to implement changes and make RAG more visible as well as approachable.

## 4.2 Suggested quality improvement initiatives

The results from the RAG were used at a Quality Improvement Workshop in April 2022 at the Internal Medicine Department for identifying areas of improvement. At the workshop in April and during the post RAG survey interviews with the Chief consultant, the following interventions were proposed–see Table 3.

A steering committee has been appointed (consisting of managers and clinicians) which would investigate how the aforementioned quality improvements should be implemented in practice. The department is planning another workshop again in autumn 2022 where the employees would work more in-depth with the implementation plan.

It is important to keep in mind that the potentials are coupled with each other and thus it is not possible to address each potential and its facets separately. For example, by minimising interruption (R7), we also improve flexibility in the ambulatory program, giving healthcare professionals more time and resources to consult with each other. According to Hollnagel [5], the Functional Resonance Analysis Model (FRAM) [42] can be used to analyse the relationship between the parts of a process and how changes in one facet can impact the others. This is a useful tool to understand and assist implementation of organisational changes.

## 4.3 Strengths and limitations

A strength of the study was its systematic process of developing and applying the RAG. This ensured that the RAG questionnaire was valid and reliable. In the process of developing the RAG, we also had external experts check the questionnaire for potential biases. Furthermore, the study included respondents from different organisational positions, which allowed us to gain a comprehensive overview at the micro and meso-level of the activities that supported the resilient performance of the internal medicine outpatient clinics. Lastly, we repeated the RAG and used the results to show how an organisational change over time could be illustrated and understood.

The study was limited to a single hospital department in Denmark, which may limit the generalisability of our results. However, it is a requirement for the RAG to be context specific and adapted to the organisation it is applied in. Furthermore, the middle managers were the least represented group in 2022. The low response rate in 2022 may be due to the staff being exposed to covid-19.

**Table 3. Overview of suggested quality improvement initiatives.**

| Resilience potentials | Suggested quality improvement initiatives |
|---|---|
| **Responding** | R1_Flexibility<br>• Implementing a joint triage of mutual patients across specialities. This would give the healthcare professionals time and flexibility to consult with colleagues.<br>• Eliminating internal referrals between the internal medicine outpatient clinics. |
| | R3_Leveraging knowledge<br>• Nurses: Ensure that at least two to three nurses know how to perform the tests, to provide redundancy in cases of unexpected situations such as illness, vacation and termination of employment. |
| | R7_Interruptions<br>• Use Integrated Communication Technology (ICT) solutions such as CETREA [41] to send messages between healthcare team members and provide digital overview of the workflow.<br>• Establishing a joint telephone number for the Internal Medicine Department with further transfer to relevant secretary and physician in the specific outpatient clinic. This would minimise irrelevant phone calls while doing work. |
| **Learning** | L5_Learning and L6_Feedback process<br>• Using the daily morning conferences to follow-up on cases and emphasise things that are going well. |

## 5. Conclusion

The RAG is a promising tool for managers to measure resilient performance and can serve as a guide to support organisational changes over time. This study validated the RAG for application in an Internal Medicine Department. The study introduced the resilient concept and enhanced the mangers and healthcare professionals' understanding the internal medicine outpatient clinics potential or ability to perform resiliently in terms of the four resilience potentials i.e. respond, monitor, learn and anticipate. The findings from the RAG indicate areas for improvement related to the potential to respond and learn. Hospitals can use the RAG to gain an understanding of the activities that support resilient performance and initiate discussion about how to maintain performance and improve the sub-indicators with the lowest scores. Involving senior managers and other key stakeholders in the process of developing and applying the RAG ensured that identified improvements were implemented and evaluated through repeating the RAG post-intervention.

## Supporting information

**S1 Appendix. Interview guide.**
(DOCX)

**S2 Appendix. RAG questionnaire original Danish version.**
(PDF)

**S3 Appendix. RAG questionnaire English version.**
(DOCX)

## Acknowledgments

Our heartfelt thank goes to Donna Lykke-Wolff for her support in data management.

## Author Contributions

**Conceptualization:** Mariam Safi, Bettina Ravnborg Thude, Frans Brandt, Robyn Clay-Williams.

**Data curation:** Mariam Safi.

**Formal analysis:** Mariam Safi.

**Investigation:** Mariam Safi.

**Methodology:** Mariam Safi, Bettina Ravnborg Thude, Robyn Clay-Williams.

**Project administration:** Mariam Safi.

**Resources:** Frans Brandt.

**Supervision:** Bettina Ravnborg Thude, Frans Brandt, Robyn Clay-Williams.

**Validation:** Mariam Safi, Bettina Ravnborg Thude, Frans Brandt, Robyn Clay-Williams.

**Visualization:** Mariam Safi.

**Writing – original draft:** Mariam Safi.

**Writing – review & editing:** Mariam Safi, Bettina Ravnborg Thude, Frans Brandt, Robyn Clay-Williams.

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
