## [Decision Letter · Decision Letter 0]

20 Jun 2022

PONE-D-22-13514The resilient potential behaviours in an Internal Medicine Department: Application of Resilience Assessment GridPLOS ONE

Dear Dr. Safi,

Thank you for submitting your manuscript to PLOS ONE. After careful consideration, we feel that it has merit but does not fully meet PLOS ONE’s publication criteria as it currently stands. Therefore, we invite you to submit a revised version of the manuscript that addresses the points raised during the review process.

We look forward to receiving your revised manuscript.

Kind regards,

Jibril Mohammed, BSc, MSc, PhD

Academic Editor

PLOS ONE

Journal Requirements:

   "This work was supported by the University Hospital of Southern Denmark as part of a Ph.D. project. "

5. Please ensure that you refer to Figure 2 in your text as, if accepted, production will need this reference to link the reader to the figure.

6. Please include a copy of Table 3 which you refer to in your text on page 22.

7. We note you have included a table to which you do not refer in the text of your manuscript. Please ensure that you refer to Table 5 in your text; if accepted, production will need this reference to link the reader to the Table.

Reviewers' comments:

Reviewer's Responses to Questions

**Comments to the Author**

1. Is the manuscript technically sound, and do the data support the conclusions?

Reviewer #1: Yes

Reviewer #2: Partly

2. Has the statistical analysis been performed appropriately and rigorously? 

Reviewer #1: Yes

Reviewer #2: No

3. Have the authors made all data underlying the findings in their manuscript fully available?

Reviewer #1: Yes

Reviewer #2: No

4. Is the manuscript presented in an intelligible fashion and written in standard English?

Reviewer #1: Yes

Reviewer #2: Yes

5. Review Comments to the Author

Reviewer #1: Thank you for sending me the manuscript entitled “The resilient potential behaviors in an Internal Medicine Department: Application of Resilience Assessment Grid” for review. The manuscript is well-written and well-organized. The following comments can help the authors to improve it.

1- In the abstract, the conclusion section needs to be revised.

2- It is better to remove Figures 1 and 2.

3- A sample of interview guideline and the questionnaire need to be presented as appendices.

4- Please add more information about the qualitative results’ trustworthiness.

5- In some parts of the manuscript the authors noted “health care professionals”. I think it is better to be more specific as for example, managers are not among healthcare professional.

6- Please present the strengths of the study first and then talk about the research limitations.

7- What will happen to the suggested quality improvement initiatives?

Reviewer #2: Dear authors,

Congratulations for the submission of paper entitled 'The resilient potential behaviours in an Internal Medicine Department: Application of Resilience Assessment Grid' to PLOS ONE. The organization resilience and its attempts to stay adaptive with challenging conditions while still maintaining/increasing the quality are critical issues nowadays. I appreciate the robust and systematic research methods including the questionnaire development. However, with the current form, I cannot recommend publication in the journal. I believe the authors should revise the paper substantially. There are some suggestions listed below for the authors to consider during revision:

1. Introduction

The elaboration on RAG and its importance within organization is quite good and easy to understand regardless the familiarity towards the topic. It is not clear to me, however, the gap analysis of this study. Why this study should be conducted? Why concerns in the internal medicine department in one teaching hospital should become concerns of the readers coming from different settings? Therefore, I recommend a stronger and clearer gap analysis elaborating what's not yet elaborated in the literature regarding this topic. Also, what are the research questions of this study? Despite the elaboration of the aim, clearer research questions are warranted to offer clear justification of the study. Further elaboration on the response or expected results of this study is also needed to give further insights on what the study would add in the literature.

2. Methods

The methods are described systematically. The setting which was explained in the introduction should be considered to be moved as the first section in the methods. The steps of instrument development are mostly well detailed. Yet, given issues in the introduction, the current stages are confusing. In addition, decisions on the method such as the exclusion of medical/nursing students, should be justified and explained. Each stage of study, including the delphi, the pilot of the instrument and so on should be better elaborated in terms of the data collection, sample, and the data analysis. I would expect better description on the thematic analysis being completed for the interview. I also wonder why there is no exploratory factor analysis or confirmatory factor analysis to support the construct of the instrument. Nor there was attempt to elaborate the reliability of the questionnaire - overall score, and subscale scores. I understand the value of reporting the mean score of each item as part of identifying the rooms for improvement in the system, however it is not clear to me why the subscale scores were not reported. Finally, at the presentation stage, I am quite confused with how was it done and how did this process add further data in this study? If so, how was it analyzed?

3. Results

The results are quite well written. Yet the clarity can be further enhanced by addressing the issues in the Methods section. All collected data be it qualitative and quantitative should be presented accurately and adequately.

4. Discussion and conclusion

The discussion is rather long. The authors may want to consider shortening the discussion section and focus more on comparing the results with the available and relevant literature as well as analyze the implications.

6. PLOS authors have the option to publish the peer review history of their article (what does this mean?). If published, this will include your full peer review and any attached files.

Reviewer #1: No

Reviewer #2: No

---

## [Author Response · Author response to Decision Letter 0]

8 Aug 2022

Please see the attached documents - cover letter and the response to reviewers.

---

## [Editor Report · Decision Letter 1]

2 Oct 2022

The resilient potential behaviours in an Internal Medicine Department: Application of Resilience Assessment Grid

PONE-D-22-13514R1

Dear Ms. Safi,

We’re pleased to inform you that your manuscript has been judged scientifically suitable for publication and will be formally accepted for publication once it meets all outstanding technical requirements.

Kind regards,

Jibril Mohammed, BSc, MSc, PhD

Academic Editor

PLOS ONE
---

## [Editor Report · Acceptance letter]

6 Oct 2022

PONE-D-22-13514R1 

The resilient potential behaviours in an Internal Medicine Department: Application of Resilience Assessment Grid 

Dear Dr. Safi:

I'm pleased to inform you that your manuscript has been deemed suitable for publication in PLOS ONE. Congratulations! Your manuscript is now with our production department. 

Kind regards, 

on behalf of

Dr. Jibril Mohammed 

Academic Editor

PLOS ONE